# Quantitative Analysis of Fluorescence Detection Using a Smartphone Camera for a PCR Chip

**DOI:** 10.3390/s21113917

**Published:** 2021-06-06

**Authors:** Jong-Dae Kim, Chan-Young Park, Yu-Seop Kim, Ji-Soo Hwang

**Affiliations:** 1School of Software, Hallym University, Chuncheon-si 24252, Korea; kimjd@hallym.ac.kr (J.-D.K.); cypark@hallym.ac.kr (C.-Y.P.); yskim01@hallym.ac.kr (Y.-S.K.); 2Bio-IT Research Center, Hallym University, Chuncheon-si 24252, Korea

**Keywords:** real-time polymerase chain reaction, PCR chip, lab-on-a-chip, threshold cycles, quantitative analysis, qPCR, microfluidic channel, fluorescence detection test

## Abstract

Most existing commercial real-time polymerase chain reaction (RT-PCR) instruments are bulky because they contain expensive fluorescent detection sensors or complex optical structures. In this paper, we propose an RT-PCR system using a camera module for smartphones that is an ultra small, high-performance and low-cost sensor for fluorescence detection. The proposed system provides stable DNA amplification. A quantitative analysis of fluorescence intensity changes shows the camera’s performance compared with that of commercial instruments. Changes in the performance between the experiments and the sets were also observed based on the threshold cycle values in a commercial RT-PCR system. The overall difference in the measured threshold cycles between the commercial system and the proposed camera was only 0.76 cycles, verifying the performance of the proposed system. The set calibration even reduced the difference to 0.41 cycles, which was less than the experimental variation in the commercial system, and there was no difference in performance.

## 1. Introduction

Real-time polymerase chain reaction (RT-PCR), also referred to as quantitative polymerase chain reaction (qPCR), is used to quantitatively measure the amount of amplification of DNA during the PCR process [1,2,3]. The RT-PCR technique has been actively implemented using PCR chips combined with lab-on-a-chip functionality that integrates a variety of technologies such as microfluidic channels [4,5]. Previously reported PCR chips were often fabricated using a variety of polymers and glasses, such as silicone and polydimethylsiloxane (PDMS), in single or hybrid forms. However, this is not cost-effective because it requires a manufacturer’s custom bonding technology to ensure fluid sealing between different materials. In addition, in some cases, the need for bulky and expensive detection equipment, and lack of standardization, are disadvantages [6,7,8,9,10,11,12,13].

The PCR chip proposed in this paper uses a printed circuit board (PCB) as a substrate. PCBs can eliminate most of the obstacles imposed by other boards in the commercialization of microfluidic devices and systems: expansion, standardization and system-level integration of microfluidic devices at a minimal cost. These benefits come from the use of well-established PCB technology that is now routinely used in the mass production of electronic circuits and consumer electronics [8]. On the bottom surface of the substrate, a heating pattern heats the microfluidic channel, and on the top surface of the substrate a thermal spread is used to uniformly heat the channel. The temperature of the chip is sensed by a thermistor attached to the bottom of the substrate, and the chip is heated or maintained at a set temperature. The proposed PCR chip consists of a polycarbonate reaction chamber, housing, and thin polypropylene film, including the PCB substrate mentioned above [14].

High-resolution CCD or CMOS cameras or photodiodes are typically used in RT-PCR instruments currently being commercialized or under development [15]. Due to the size of the cameras, commercial products are bulky, expensive and take up a lot of space in the laboratory [16]. Although commercial instruments using photodiodes are inexpensive and use smaller components than cameras, they still require various optical components such as lenses, pinholes, dichroic mirrors and prisms, and it is difficult to minimize the scale of the system without redesigning the complicated optical structures [6,17,18,19].

Our team proposed an RT-PCR chip system using a DSLR camera as a fluorescence detection sensor in previous studies [8,9]. This system not only enables successful DNA amplification and fluorescence detection, but also allows the amplification process to be monitored with photographs, enabling the detection and resolution of experimental errors and system errors. However, the DSLR camera is not a suitable sensor for use in a small system.

The open platform, which has been rapidly developing in recent years, is being applied in various ways to embedded systems. Open platform means providing tools that enable users to use functions, information and services in a desired way. In addition, there is an advantage in that it is possible to easily manufacture various systems in which an IoT system is embedded due to communication scalability. High-quality smartphone camera modules are being developed based on an open platform. Therefore, if an open platform is used, it is easy to develop a system that can be accessed anytime, anywhere [20,21,22].

Furthermore, advances in smartphones have enabled the replacement of the smartphone’s camera module with a fluorescence detection sensor, since the ultra-small and high-quality camera module used in the smartphone can be easily obtained at a low cost [16,23,24]. 

In this paper, we propose a small RT-PCR system that illuminates blue LEDs on a PCR chip diagonally and detects fluorescence using a miniature surveillance camera module with a CMOS sensor for smartphones. DNA amplification and fluorescence detection were successfully performed using the proposed system [16,23]. We also propose a method to optimize the parameter for the threshold cycle, which is an important measure of quantitative analysis, and to compare the threshold cycles obtained with the optimized parameter to those of the reference commercial system. Experimental results on three different concentrations of a DNA reagent showed that the difference between the threshold cycles measured by the proposed system and those from the reference commercial system was only 0.41 cycles.

## 2. The Proposed Smartphone Camera-Based RT-PCR System

There have been many reports on fluorescence detection using smartphones, but most of them have used smartphones directly. As reported by Gou et al., it was also possible to directly detect fluorescence using a smartphone. Since it was difficult to develop and maintain analysis programs that require expertise such as *C_t_* (threshold cycle), melting curve, etc., we implemented a system using an open platform camera that can be connected to a personal computer, which is relatively easy to develop and implement.

In addition, for temperature control of the qPCR process, a Peltier and a commercial thermal cycler using a bulky fan are often used for cooling, which leads to some restrictions on miniaturizing the system [25,26,27].

However, in the proposed qPCR system, fluorescence detection is performed with an open platform camera module. A heating pattern of a PCB embedded in a chip and a small fan is used for temperature control. As shown in Figure 1, the chip and fan (including chip connector) responsible for heating and cooling in the proposed system can be configured within 80 mm (W) × 10 mm (L) × 40 mm (H). If the local system including this and the optics (camera module) are all configured, a prototype can be implemented within dimensions of 80 mm (W) × 120 mm (L) × 120 mm (H). In addition, since the area of the heating pattern is small, it allows increasing the temperature in a short time.

There have been various studies involving fabrication a PCR chip using a microfluidic channel spin-coated with polyethylene after constructing a chamber using PDMS, [28].

In this study, the PCR chip consists of a PCB substrate and a microfluidic channel made of polycarbonate. The chamber is covered with polypropylene tape, and the PCB under the chamber contains a heating pattern, which heats the chamber. The manufacturing process is simple because the chips are manufactured using readily available tape.

Due to the well-developed PCB manufacturing environment, mass production is easy, and the PCB price per chip is expected to be within $ 0.1.

Figure 2 shows a block diagram of the proposed system. The PCR system consists of two parts: a system for driving the PCR chip, and a fluorescence detection system. The chip driving system includes a microcontroller (PIC 18F4550, Microchip Technology Inc., Chandler, AZ, USA) that has an analog-to-digital converter (ADC), pulse-width modulation (PWM) and a USB interface. The heater and fan are controlled by PWM and driven with field-effect transistors. The thermistor resistance in the PCR chip is changed to a digital temperature value derived from the voltage divider and ADC. Since the NTC thermistor used as a temperature sensor has poor linearity, a voltage divider was made by connecting a precision reference resistor and a precision resistor in series to construct a straightening circuit. A thermistor resistor was used as a divider resistor to compensate for the shortcomings of the temperature sensor. Every 5 ms, the temperature is calculated in the local system and sent to the host through the USB interface. The host determines the PWM value for the proportional-integral-differential control and sends it back to the local system. Temperature control is accomplished using a heater and a fan (BFB0405HHA-A, Delta Electronics, Taipei, Taiwan) to heat or cool. In the proposed system, the PCR chip was designed for a heating and cooling rate of about 10 °C/s with a temperature error of 0.5 °C. Therefore, the temperature must be sampled at about 10 times in 50 ms, during which the temperature changed by 0.5 °C. Because it was difficult to set this resolution with the standard timer provided by MS Windows, we set it to less than 5 ms using a high-precision event timer (HPET). The advantage of this host-local architecture is a reduction in the overall cost [29,30,31].

The PCR step requires a preset operation to allow repeated operation at a set temperature for a predetermined time during the PCR process [32]. Generally, the PCR step consists of denaturing DNA or RNA at 94 °C–95 °C, annealing the primer at 58 °C–60 °C, and elongating the DNA or RNA using primers at 72 °C–73 °C. The functions related to the execution and file management of the set PCR step file are processed by the host through the file input/output process in the graphical user interface environment [29,30,31].

In the proposed system, we used an ultra-high-resolution and small surveillance camera module for fluorescence detection. The module has a high-performance CMOS sensor (Sony IMX179, Tokyo, Japan), which is employed for various smart devices including smartphones. To detect fluorescein amidite fluorescence (FAM), a blue LED with a brightness of 9600 mcd (product number: 516-2275-1-ND, Broadcom Limited, San Jose, CA, USA) was illuminated diagonally on the chip. We fabricated a compact RT-PCR system, as shown in Figure 1, by arranging an excitation filter (466 nm CWL, 40 nm bandwidth, OD 6 fluorescence filter, interference filter, Edmund Optics, Barrington, NJ, USA) and emission filter (525 nm CWL, 45 nm bandwidth, OD 6 fluorescence filter, interference filter, Edmund Optics) to detect the wavelength of specific fluorescence in front of the camera [16,23].

## 3. PCR Chip

The PCR chip includes a heater pattern and a thermistor at the bottom of a 200 µm thick PCB substrate, and a rectangular copper pattern at the top for uniform thermal spreading, as shown in Figure 3. Polycarbonate material was used to construct the housing and the reaction chamber pit, and it was sealed with single-sided adhesive polypropylene film. The PCB substrate was mechanically attached tightly to the sealing film by welding the bottom cover to the top cover housing. The PCB substrate was finished in matte black to prevent light reflection, and the thermal spreader was coated with tin to make it easier to distinguish changes in fluorescent brightness [33]. In our previous version, the thermal spreader was silk-printed with white ink; however, the tin coating delivers a better signal-to-noise ratio than silk [29,31].

Figure 3c shows the chip where the PCR reaction is completed. The reagent is injected through the inlet on the left. Since the inlet is directly connected to the other device and does not have a separate lid, it is sealed using a sterile tip and glue gun. Air bubbles generated during the PCR reaction reduce the accuracy of fluorescence detection. Bubbles are generated when the chip proposed in this paper is used, but the bubble generation problem can be solved to some extent because bubbles are collected in the cone-shaped part of the upper part.

## 4. Experiment Methods

### 4.1. RT-PCR Process

We prepared *Chlamydia trachomatis* (CT) DNA at a concentration of 10^−1^ ng/μL, with 10-fold and 100-fold dilutions, before performing PCR. The reagents in the experiment used a mixture of 4.5 μL CT DNA, 15 μL TaqMan™ Gene Expression Master Mix, 7.5 μL primer mixture (primer F/R, probe), and 3 μL distilled water; 30 μL of this mixed reagent was injected into the PCR chip. For quantitative analysis of RT-PCR, mixed reagents containing 10^−1^ ng/μL of DNA were injected into two chips in sequence and inserted into two prepared RT-PCR systems (proposed system). We then performed DNA amplification and fluorescence detection. The same experiment was repeated with DNA at 10^−2^ ng/μL and 10^−3^ ng/μL concentrations.

The PCR step involved preincubation at 50 °C for 2 min, preheating at 95 °C for 10 min, denaturation at 95 °C for 15 s, and then annealing at 60 °C for 1 min. During this process, the denaturation step and the annealing step were repeated 40 times.

The fluorescence detection process was performed in the following manner. First, the LED was turned on 2 s before the end of the 60 °C annealing section of the cycle. A picture was taken 1 s before the end of this annealing section, and the LED was turned off at the end of this section. This process was repeated every cycle. As a result, 40 images were captured after the PCR process was finished. Using two systems, we performed four experiments on three concentrations for each system.

### 4.2. Result Image Processing

The relative fluorescence intensity was calculated through image processing as follows. The area corresponding to the reaction chamber was set to the region of interest (ROI), as shown in Figure 4, and converted to grayscale. An average value of the ROI was taken as the relative fluorescence intensity. Figure 5 shows the RT-PCR result images obtained during the amplification process of CT DNA at a concentration of 10^−1^ ng/μL. The fluorescence intensity changed from the first cycle to the last cycle, demonstrating the status of DNA amplification.

The relative fluorescence intensity was normalized by subtracting the average value of the initial 10 cycles. This eliminates the fluorescence deviation caused by variations in the master mix or optics before amplification [34]. Figure 6 shows the normalized plot of experimental results for the amplification with initial concentrations of 10^−1^ ng/μL, 10^−2^ ng/μL, and 10^−3^ ng/μL in two individual sets. As shown in the figure, the two sets showed different intensity gaps: the set with a larger intensity gap is referred to as A (blue lines in Figure 6), and the other as B (orange lines). The normalized data were logarithmically transformed to compute the threshold cycle (*C_t_*) at which the fluorescence intensity significantly increased above the background fluorescence. Figure 7 shows the logarithmic change in the first experimental result for two sets of three concentrations.

To obtain the parameter for the *C_t_* values of the proposed system, we referred to *C_t_* values from a commercial RT-PCR instrument. The *C_t_* values from the reference and the proposed system are expressed as vectors represented by C and $\hat{C}$, and the parameter *t* is determined by minimizing the distance according to the following equation: (1)$$d\left(s,e,t\right)=\|C-\hat{C}\left(s,e,t\right)\|,$$
where $s,e,t,C_{t},{\hat{C}}_{t}$ are the set-id, the experimental id, the parameter to be determined, the reference *C_t_* vector from the proposed system and the *C_t_* vector obtained from the proposed system, respectively. The distance between the reference *C_t_* value and the estimated *C_t_* vector is a function of the set-id and experiment id. If the distance for the individual experiment is similar, calibration between experiments is not necessary once a parameter *t* is obtained that minimizes the distance for all experiments by Equation (2):(2)$$d_{e}\left(s,t\right)=\sum_{e}\left\|C-\hat{C}\left(s,e,t\right)\right\|$$

Likewise, if the optimum parameter *t* is similar between sets A and B, then *t* can be obtained by minimizing the following Equation (3) and applying it to both sets without calibration:(3)$$d_{a}\left(t\right)=\sum_{s,e}\left\|C-\hat{C}\left(s,e,t\right)\right\|$$

On the other hand, the intensity gap for set B is multiplied by the intensity ratio of the amplification ends of sets A and B to correct set B’s intensity gap, and the curve for set B is increased to match the intensity of set B to the end of set A. This calibration allows for a smaller distance between the *C_t_* values obtained from the proposed system and those of the reference RT-PCR system.

## 5. Experimental Results

Figure 8 shows the distance according to the threshold for each experiment for set A. As shown in the figure, the distance is less than 0.5 when the threshold is below 3.0, despite fluctuations in the parameter with minimum distance. Similar results were observed in set B, and Table 1 shows the optimal threshold for each set and the distance between the reference *C_t_* value and the calculated *Ct*. This indicates the advantage of obtaining and applying a threshold that minimizes the distance $d_{e}\left(s,t\right)$ for all experiments. Table 2 shows the thresholds and the distance at which the total distance $d_{e}\left(s,t\right)$ for all experiments is minimized in each set. The last row of Table 2 (marked ‘A + B’) shows the result of minimizing $d_{a}\left(t\right)$, which is the sum of the distance $d_{e}\left(s,t\right)$ for all sets. While the distance of each set was 0.3 and 0.34 cycles, respectively, as shown in Figure 9, the distance was 0.78 cycles when minimizing $d_{a}\left(t\right)$, as shown in the last row of Table 2. This is still similar to the experimental standard deviation (0.73) of the reference system.

The average intensity between cycles 35 and 40 was obtained for calibration, where the average intensity of A and B was 74.2 and 41.2, respectively. Therefore, the brightness of set B was multiplied by the ratio of A and B (1.8). Figure 10 shows the logarithmic transformation of the average fluorescence intensity after calibration. Unlike Figure 7, which was not calibrated, the same level of log intensity was displayed for both sets at the end of the amplification when calibrated.

In Figure 11, the log threshold with the minimum distance for all sets A and B after calibration was 2.3, and the distance value at this time was 0.4 cycles, which was 50% less than that before calibration.

When estimating the initial concentration using *Ct*, the difference from one cycle indicates a two-fold difference in the initial concentration. Inducing from the 0.78 cycle error in the proposed system, it is possible to predict the initial concentration within 74% (2^0.8^–1) of the error. The error can be further decreased through the proposed calibration to within 32% (0.4 cycles). Given that the standard deviation of the commercial equipment (ABI) was 0.73, the proposed system showed similar performance.

## 6. Conclusions

Many existing commercial RT-PCR instruments are bulky or expensive as they contain expensive fluorescence detection devices or are composed of complex optical structures. In the proposed system, the spatial limitations of the commercial equipment are addressed by implementing an inexpensive, ultra-compact and high-quality smartphone camera module as a fluorescence detection sensor. To use the proposed system for quantitative analysis, this work presents a method for obtaining optimal thresholds and parameters by referring to a commercial RT-PCR instrument. Applying the parameters obtained by the proposed method, the difference between the threshold cycle obtained from the proposed system and that for a commercial RT-PCR instrument is only about 0.78 cycles, which can be further reduced to 0.4 cycles by calibration. Considering that the standard deviation is about 0.73 cycles in commercial equipment, the proposed system demonstrates similar performance to that of commercial equipment.

## Figures and Tables

**Figure 1 sensors-21-03917-f001:**
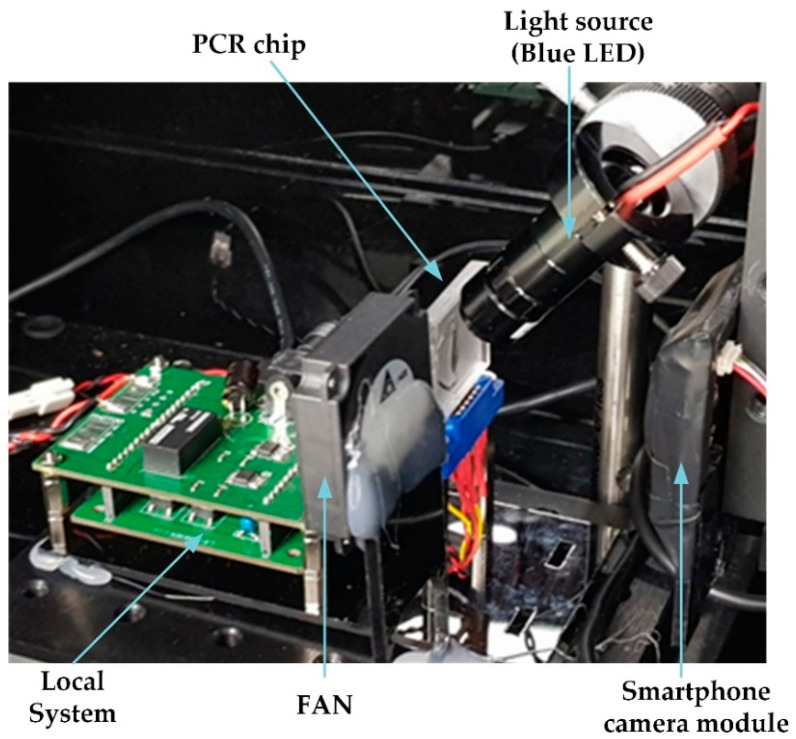
System setup of the proposed RT-PCR system.

**Figure 2 sensors-21-03917-f002:**
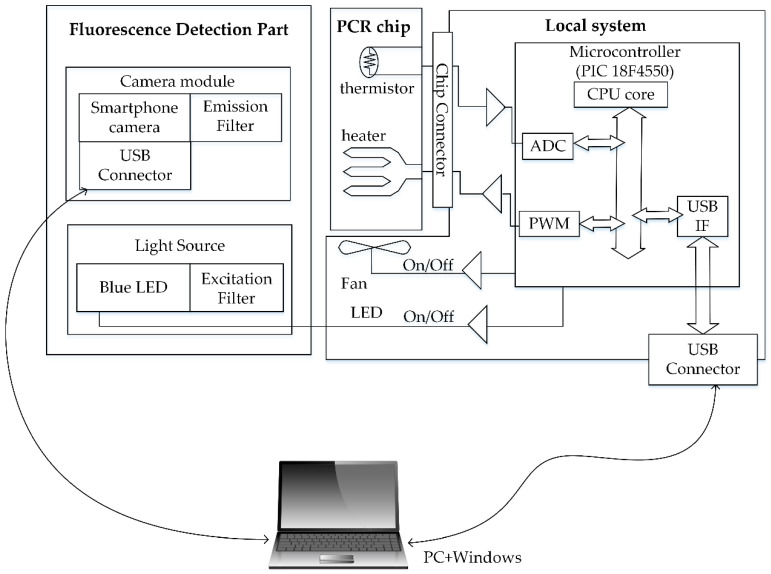
Block diagram of proposed RT-PCR system.

**Figure 3 sensors-21-03917-f003:**
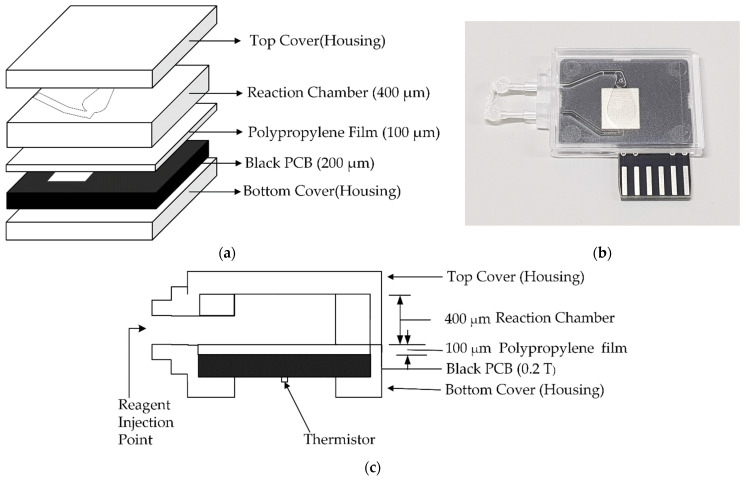
PCR chip structure: (**a**) assembly diagram, (**b**) PCR chip used in the experiment, and (**c**) cross-section.

**Figure 4 sensors-21-03917-f004:**
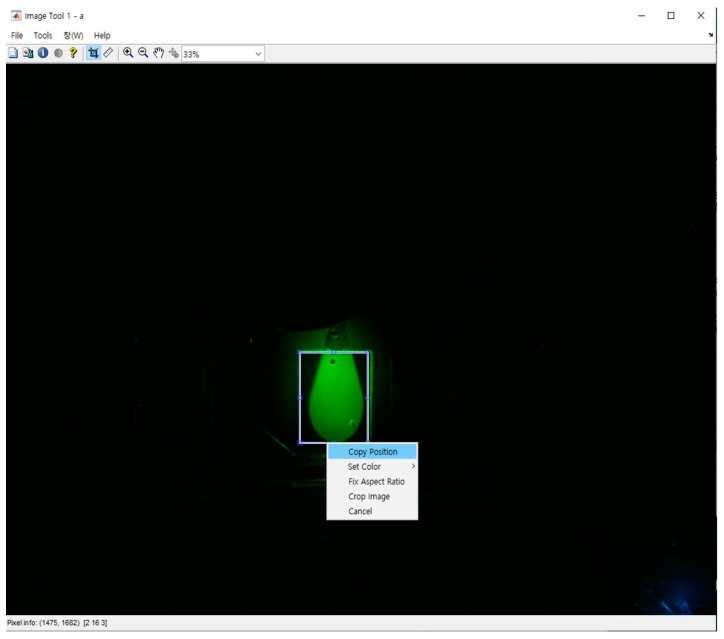
Region of interest (ROI) selection for image processing.

**Figure 5 sensors-21-03917-f005:**
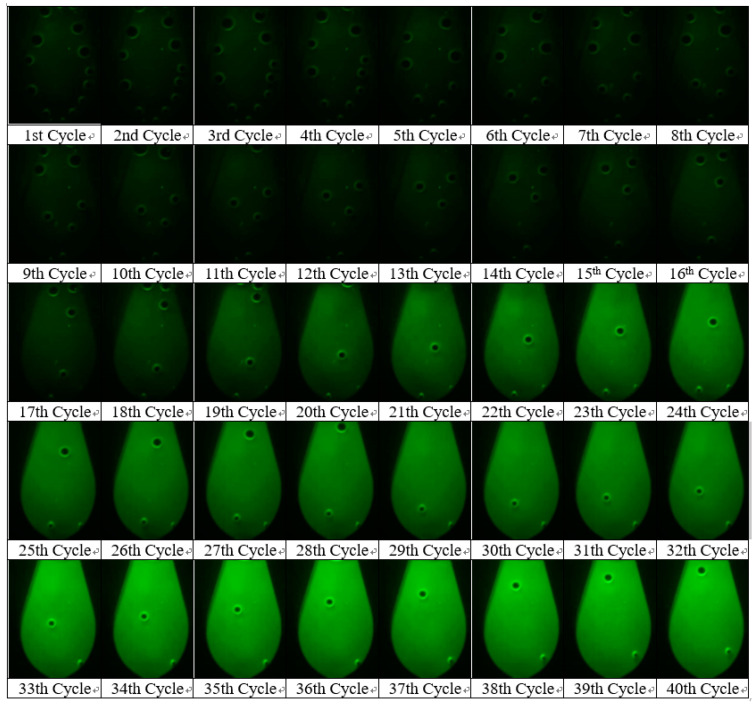
RT-PCR result images of CT DNA obtained every PCR cycle at a concentration of 0.1 ng/μL.

**Figure 6 sensors-21-03917-f006:**
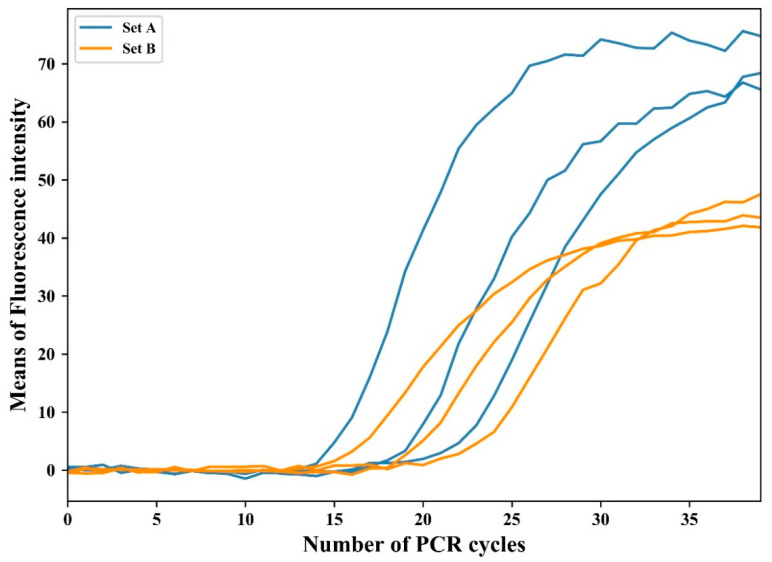
Fluorescence intensity change in the first experiment after baseline calibration.

**Figure 7 sensors-21-03917-f007:**
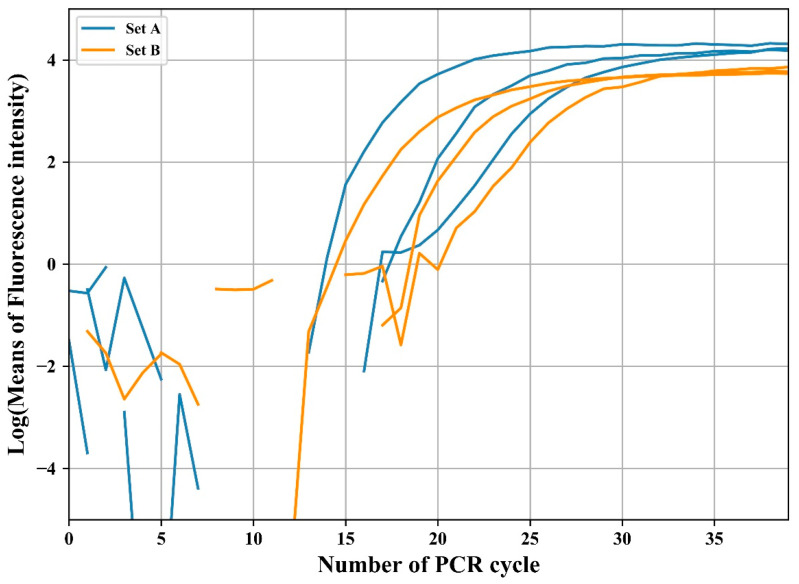
Logarithmic transformation of fluorescence intensity for threshold cycle determination.

**Figure 8 sensors-21-03917-f008:**
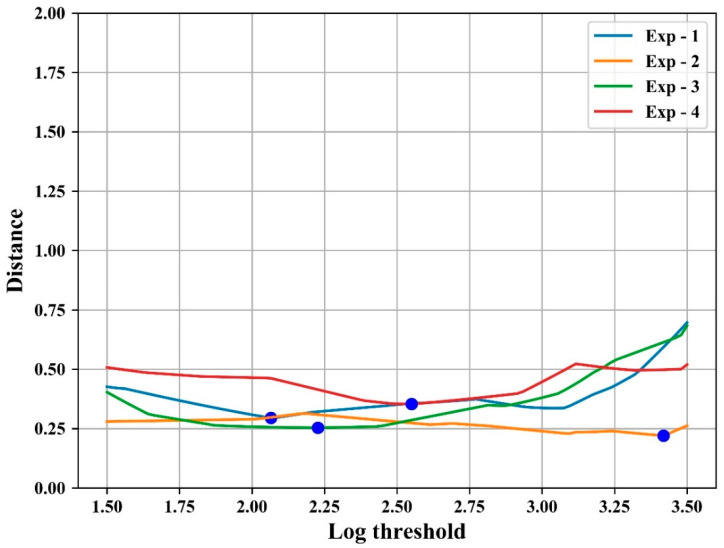
Distance according to log threshold for each experiment in set A.

**Figure 9 sensors-21-03917-f009:**
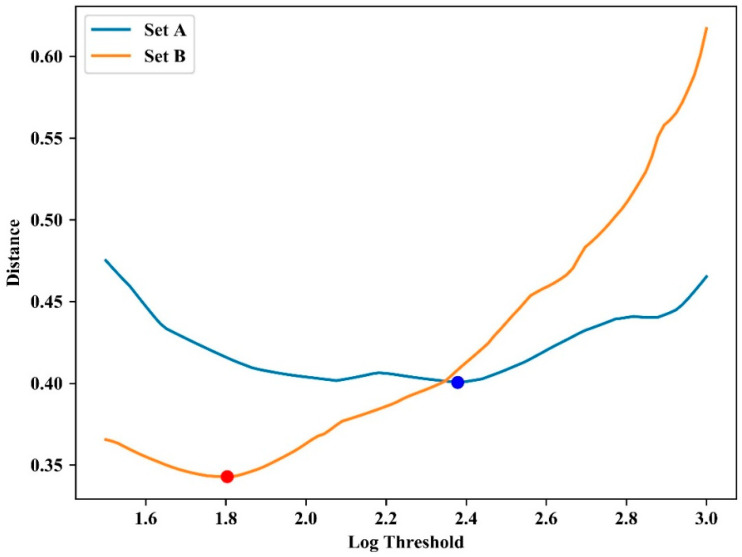
Distance distribution according to log threshold for each experiment set.

**Figure 10 sensors-21-03917-f010:**
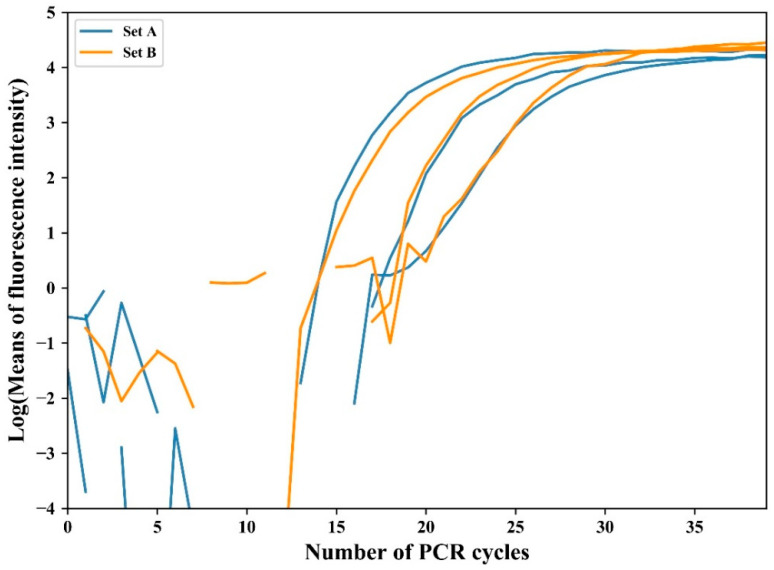
Logarithmic transformation of the fluorescence intensity of the entire experiment after the calibration process.

**Figure 11 sensors-21-03917-f011:**
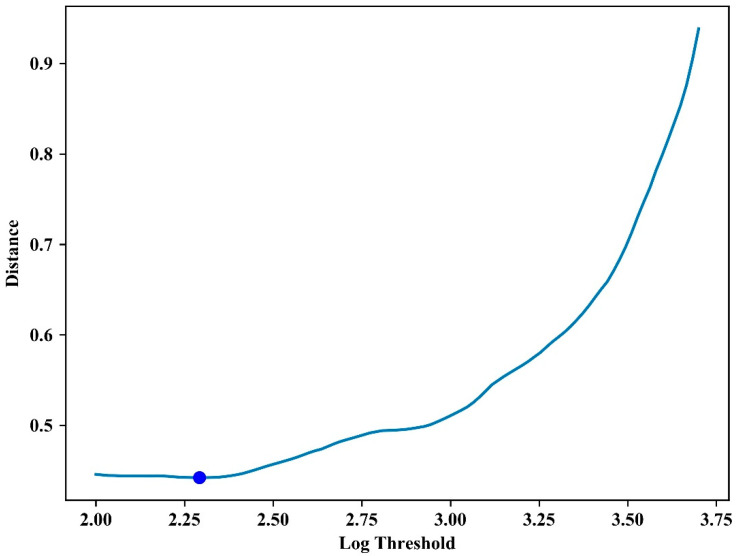
Distance distribution according to the log threshold for all experiments after calibration.

**Table 1 sensors-21-03917-t001:** Distances of *C_t_* values according to the threshold of each experiment.

Exp No.	Set A		Set B	
	Threshold	Distance	Threshold	Distance
1	2.06	0.41	1.88	0.01
2	3.41	0.09	0.5	0.14
3	1.88	0.37	1.76	0.01
4	2.57	0.33	2.04	0.33

**Table 2 sensors-21-03917-t002:** Log thresholds and distances when integrating experiments for each set.

Exp.-Set	Threshold	Distance
A	2.37	0.4
B	1.80	0.34
A + B	1.5	0.78

## Data Availability

Not applicable.

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
