# Peer review of "Quantitative Analysis of Fluorescence Detection Using a Smartphone Camera for a PCR Chip"

_sensors, 2021, doi:10.3390/s21113917_

Round 1
Reviewer 1 Report
The authors report a qPCR platform that integrates home-made thermal cycling and smartphone camera based imaging. However, this work seems to have minimal novelty, since similar systems have been published. Sensors and Actuators B: Chemical 298 (2019): 126782, arXiv preprint arXiv:1606.02252 (2016), Biosensors and Bioelectronics 120 (2018): 144-152 all reported integrated systems like this paper. Moreover, Angewandte Chemie 132.27 (2020): 11074-11081 reported an integrated qPCR microfluidic chip and a commercial thermal cycler that enabled point-of-care diagnostics, and Biosensors and Bioelectronics 126 (2019): 725-733 reported a laser-based thermal cycling system, which is in contrast to what the authors claimed in the introduction ("However, this type of system is not suitable for small 32 point-of-care systems due to its use of a large thermal cycling system"). The authors must discuss the pros and cons of existing works and justify the novelty of this work.
The authors did not describe the microfluidic chip in detail. There was no magnified view of the chip. And how was the PCR reagent injected into the chip, and how was the chip sealed? Were there bubble generation issues during thermal cycling?
Figure 2 did not clearly show each component. Arrows are recommended to specify each component. Add multi-angle views and magnified views if possible to better illustrate the system.
The model numbers of the microcontroller and the fan were not described. The "voltage divider" was not explained.
Author Response
The authors report a qPCR platform that integrates homemade thermal cycling and smartphone camera-based imaging. However, this work seems to have minimal novelty, since similar systems have been published. Sensors and Actuators B: Chemical 298 (2019): 126782, arXiv preprint arXiv:1606.02252 (2016), Biosensors and Bioelectronics 120 (2018): 144-152 all reported integrated systems like this paper. Moreover, Angewandte Chemie 132.27 (2020): 11074-11081 reported an integrated qPCR microfluidic chip and a commercial thermal cycler that enabled point-of-care diagnostics, and Biosensors and Bioelectronics 126 (2019): 725-733 reported a laser-based thermal cycling system, which is in contrast to what the authors claimed in the introduction ("However, this type of system is not suitable for small 32 point-of-care systems due to its use of a large thermal cycling system"). The authors must discuss the pros and cons of existing works and justify the novelty of this work.
[Response to Reviewer 1] : Explanation have been added as recommended
The authors did not describe the microfluidic chip in detail. There was no magnified view of the chip. And how was the PCR reagent injected into the chip, and how was the chip sealed? Were there bubble generation issues during thermal cycling?
[Response to Reviewer 1] : corrections have been done and added explanation as recommended
Figure 2 did not clearly show each component. Arrows are recommended to specify each component. Add multi-angle views and magnified views if possible to better illustrate the system.
[Response to Reviewer 1] : Corrections have been done as recommended.
The model numbers of the microcontroller and the fan were not described. The "voltage divider" was not explained.
[Response to Reviewer 1] : Explanation have been added as recommended

Reviewer 2 Report
The quantative analysis of the RT-PCR chip compact system using flurescence detection sensor is introduced. The proposed RT-PCR system with blue LEDs printed on a PCR chip using surveillance camera module is interesting. The method of the threshold cycle of the quantitative analysis with the optimized parameter is necessary to be commercialized. Therefore, the proposed approach is worthwhile to be published. Authors showed the detail architecture of the proposed PCR system. However, authors need to add some references because there are small numbers of the references in the entire manuscript. In addition, Figure qualities need to be improved before acceptance. Therefore, the manuscript could be minor revision if authors follows the comments.
1. In the reference section, authors should use abbreviated journal names.
2. Please add the reference (High-resolution CCD or CMOS cameras or photodiodes are typically used in RT-PCR instruments that are currently being commercialized or are under development) with the reference (Rivet, Catherine, et al. "Microfluidics for medical diagnostics and biosensors." Chemical Engineering Science 66.7 (2011): 1490-1507.) or another reference.
3. Please add the reference (Due to the size of the cameras, commercial products are bulky, expensive, and take up a lot of ~) with the reference (Hwang, Ji-Soo, et al. "Real-time Polymerase Chain Reaction System Using a Camera for Open Platform." Sensors and Materials 31.5 (2019): 1647-1655.) or another reference.
4. Please add the reference (This eliminates the fluorescence deviation caused by variations in the master mix or optics before amplification) with the reference (Kim, Hanyoup, et al. "Nanodroplet real-time PCR system with laser assisted heating." Optics express 17.1 (2009): 218-227. ) or another reference.
5. Please add the reference (However, the DSLR camera is not a suitable sensor for use in a small system) with the reference (Choi, H.; Choe, S.-w.; Ryu, J. Optical Design of a Novel Collimator System with a Variable Virtual-Object Distance for an Inspection Instrument of Mobile Phone Camera Optics. Appl. Sci. 2021, 11, 3350. ).
6. Please improve the Figure 1 quality because the block diagram of Figure 1 is not clearly to be seen.
7. Please delete Figure 3 comments in Lines 123-125.
8. Please increase Figures 3 and 6~11 font sizes.
9. In Line 123, please change PCR chip structure: (a) assembly diagram, (b) cross section to PCR chip structure: (a) assembly diagram and (b) cross section.
10. In Line 184, please change equatin (2) to Equation (2).
11. Please add the reference (The PCR step requires a preset operation to allow repeated operation at a set temperature for a predetermined time during the PCR process ) with the reference (Turechek, W. W., J. S. Hartung, and Jennifer McCallister. "Development and optimization of a real-time detection assay for Xanthomonas fragariae in strawberry crown tissue with receiver operating characteristic curve analysis." Phytopathology 98.3 (2008): 359-368. ) or another reference.
12. please change "Figure 5 shows RT-PCR result images obtained" to " Figure 5 shows the RT-PCR result images obtained".
Author Response
The quantitative analysis of the RT-PCR chip compact system using fluorescence detection sensor is introduced. The proposed RT-PCR system with blue LEDs printed on a PCR chip using surveillance camera module is interesting. The method of the threshold cycle of the quantitative analysis with the optimized parameter is necessary to be commercialized. Therefore, the proposed approach is worthwhile to be published. Authors showed the detail architecture of the proposed PCR system. However, authors need to add some references because there are small numbers of the references in the entire manuscript. In addition, Figure qualities need to be improved before acceptance. Therefore, the manuscript could be minor revision if the authors follows the comments.
- In the reference section, authors should use abbreviated journal names.
[Response to Reviewer 2] : Corrections have been done as recommended. - Please add the reference (High-resolution CCD or CMOS cameras or photodiodes are typically used in RT-PCR instruments that are currently being commercialized or are under development) with the reference (Rivet, Catherine, et al. "Microfluidics for medical diagnostics and biosensors." Chemical Engineering Science 66.7 (2011): 1490-1507.) or another reference.
[Response to Reviewer 2] : The reference has been added as recommended. - Please add the reference (Due to the size of the cameras, commercial products are bulky, expensive, and take up a lot of ~) with the reference (Hwang, Ji-Soo, et al. "Real-time Polymerase Chain Reaction System Using a Camera for Open Platform." Sensors and Materials 31.5 (2019): 1647-1655.) or another reference.
[Response to Reviewer 2] : The reference has been added as recommended. - Please add the reference (This eliminates the fluorescence deviation caused by variations in the master mix or optics before amplification) with the reference (Kim, Hanyoup, et al. "Nanodroplet real-time PCR system with laser-assisted heating." Optics express 17.1 (2009): 218-227. ) or another reference.
[Response to Reviewer 2] : The reference has been added as recommended. - Please add the reference (However, the DSLR camera is not a suitable sensor for use in a small system) with the reference (Choi, H.; Choe, S.-w.; Ryu, J. Optical Design of a Novel Collimator System with a Variable Virtual-Object Distance for an Inspection Instrument of Mobile Phone Camera Optics. Appl. Sci. 2021, 11, 3350. ).
[Response to Reviewer 2] : The reference has been added as recommended. - Please improve the Figure 1 quality because the block diagram of Figure 1 is not clear to be seen.
[Response to Reviewer 2] : The figure has been modified as recommended. - Please delete Figure 3 comments in Lines 123-125.
[Response to Reviewer 2] : The comments have been deleted as recommended. - Please increase Figures 3 and 6~11 font sizes.
[Response to Reviewer 2] : The figure has been modified as recommended. - In Line 123, please change PCR chip structure: (a) assembly diagram, (b) cross section to PCR chip structure: (a) assembly diagram and (b) cross section.
[Response to Reviewer 2] : The caption has been corrected as recommended. - In Line 184, please change equation (2) to Equation (2).
[Response to Reviewer 2] : Corrections have been added as recommended. - Please add the reference (The PCR step requires a preset operation to allow repeated operation at a set temperature for a predetermined time during the PCR process ) with the reference (Turechek, W. W., J. S. Hartung, and Jennifer McCallister. "Development and optimization of a real-time detection assay for Xanthomonas fragariae in strawberry crown tissue with receiver operating characteristic curve analysis." Phytopathology 98.3 (2008): 359-368. ) or another reference.
[Response to Reviewer 2] : The reference has been added as recommended. - please change "Figure 5 shows RT-PCR result images obtained" to " Figure 5 shows the RT-PCR result images obtained".
[Response to Reviewer 2] : Corrections have been added as recommended.
Reviewer 3 Report
The authors base their technical proposal on the premise: most commercial real-time PCR instruments are bulky, with expensive fluorescent signal detection sensors or complex optical structures. They propose a simple system that uses a smartphone camera module as an ultra-small, high-performance, low-cost sensor for fluorescence detection. A quantitative analysis of changes in fluorescent signal intensity showed good camera performance compared to commercial systems. They also analyzed performance in threshold cycle (Ct)-based experiments using a commercial real-time PCR system. They report an overall difference in Ct values of 0.76 cycles. In consideration of the results obtained, they conclude that the proposed system shows performance similar to that observed in commercial equipment.
The introduction shows a clear scientific basis, but the hypothesis /premise is quite simple. The experimental design and methodology are standard, with a limited contribution to the field. A thorough revision of the hypothesis is suggested, with a rigorous observation of the scientific method. Although they made a reasonable observation, the primary research shows poor originality. Finally, a thorough review of the English language is advised, preferably using a professional proofreading service that employs native speakers with scientific training.
Given the above, publication in “Sensors” is recommended after submitting an improved version of the manuscript.
Author Response
The authors base their technical proposal on the premise: most commercial real-time PCR instruments are bulky, with expensive fluorescent signal detection sensors or complex optical structures. They propose a simple system that uses a smartphone camera module as an ultra-small, high-performance, low-cost sensor for fluorescence detection. A quantitative analysis of changes in fluorescent signal intensity showed good camera performance compared to commercial systems. They also analyzed performance in threshold cycle (Ct)-based experiments using a commercial real-time PCR system. They report an overall difference in Ct values of 0.76 cycles. In consideration of the results obtained, they conclude that the proposed system shows performance similar to that observed in commercial equipment.
The introduction shows a clear scientific basis, but the hypothesis /premise is quite simple. The experimental design and methodology are standard, with a limited contribution to the field. A thorough revision of the hypothesis is suggested, with a rigorous observation of the scientific method. Although they made a reasonable observation, the primary research shows poor originality. Finally, a thorough review of the English language is advised, preferably using a professional proofreading service that employs native speakers with scientific training.
Given the above, publication in “Sensors” is recommended after submitting an improved version of the manuscript.
[Response to Reviewer 3] : The proposal system is the size of the entire system, including the fluorescence detection system and chip, is small, and low-cost constituent materials are used to manufacture disposable chips, and the manufacturing process is simple, which is advantageous for mass production.
Compared to the previously reported systems, the temperature control unit and the fluorescence detection part can be manufactured in a smaller size, so it is advantageous for commercialization, and it is believed that it can help a lot in the field diagnosis field.
We are collecting more experimental result data than the currently analyzed data and analyzing the results, and we are also analyzing according to temperature control, which is an important factor in the amplification results. In addition, we are currently trying to analyze clinical data in addition to analyzing standard substances.
Finally, we are reviewing and verifying the English proofreading again according to the reviewer's recommendation.

Round 2
Reviewer 1 Report
The authors failed to address my concerns regarding the novelty of this work. They made wrong claims when comparing their platform with existing ones.
- Section 2: "There have been many reports on fluorescence detection using smartphones, but most of them have used smartphones directly. ...However, in the proposed qPCR system, fluorescence detection is performed with an open platform camera module." What is the advantage of using "an open platform camera module"? A laptop is required for this work, but existing works such as Gou et al. ("Smartphone-based mobile digital PCR device for DNA quantitative analysis with high accuracy." Biosensors and Bioelectronics 120 (2018): 144-152.) only need a smart phone.
- Section 2: "In addition, for temperature control of the 70 qPCR process, a peltier and a commercial thermal cycler using a bulky fan were often used to cool it. In this case, it seems that there will be some restrictions on miniaturizing the system....heating pattern of a PCB embedded in a chip and a small fan is used for temperature control. Therefore, in the case of manufacturing the entire system, it was possible to produce a prototype within 120mm x 80mm x 120mm. "
Introduction: "Most of the RT-PCR systems containing this type of PCR chip were configured to heat the PCR chip externally. However, this type of system is not suitable for small point-of-care systems due to its use of a large thermal cycling system "
Wrong claims. Gou et al. used a turbo fan (C81H Dayu, PCCOOLER, Shenzhen, China) of 90.5 (L) x 90.5 (W) x29 (H) mm. Jiang et al. used a thermoelectric cooler (TEC; ATE1-TC-127-8ASH, Analog Technologies) of 40 mm x 40 mm. In addition, Ouyang et al. ("One‐Step Nucleic Acid Purification and Noise‐Resistant Polymerase Chain Reaction by Electrokinetic Concentration for Ultralow‐Abundance Nucleic Acid Detection." Angewandte Chemie 132.27 (2020): 11074-11081.) used MiniPCR mini8 with a fan of approximately 40 mm x 40 mm. The authors used a fan (BFB0405HHA-A) of approximately 40 mmx40 mm. - Section 2: "Furthermore, in the case of reported PCR chips, most of the cases are heated outside the chip. The difference between these is that the proposed chip includes a heating part in the chip so that the temperature control part can be downsized. " The authors attaches a PCB to the bottom of the PCR chip for heating. This is not much different from other systems where the heating pad is at the bottom of the PCR chip. In addition, in the authors' design, they need a new PCB for each PCR chip, which increases the cost. The area of the heating part is determined by the area of the reaction chamber, not necessarily by how the heating part is integrated. It is unclear how the authors' design may make the overall system smaller.The authors need to thoroughly re-structure the introduction so that readers understand what is new in this work.
